# Cathepsin K in Pathological Conditions and New Therapeutic and Diagnostic Perspectives

**DOI:** 10.3390/ijms232213762

**Published:** 2022-11-09

**Authors:** Olja Mijanović, Aleksandra Jakovleva, Ana Branković, Kristina Zdravkova, Milena Pualic, Tatiana A. Belozerskaya, Angelina I. Nikitkina, Alessandro Parodi, Andrey A. Zamyatnin

**Affiliations:** 1Dia-M, LCC, 7 b.3 Magadanskaya Str., 129345 Moscow, Russia; 2The Human Pathology Department, Sechenov First Moscow State University, 119991 Moscow, Russia; 3HoxLife Science GmbH, Gutleutstraße 169-171, 60327 Frankfurt am Main, Germany; 4Department of Forensics Engineering, University of Criminal Investigation and Police Studies, Cara Dusana 196, 11000 Belgrade, Serbia; 5AD Alkaloid Skopje, Boulevar Alexander the Great 12, 1000 Skopje, North Macedonia; 6Institute Cardiovascular Diseases Dedinje, Heroja Milana Tepica 1, 11000 Belgrade, Serbia; 7Bach Institute of Biochemistry, Research Center of Biotechnology, Russian Academy of Sciences, 119071 Moscow, Russia; 8ArhiMed Clinique for New Medical Technologies, Vavilova St. 68/2, 119261 Moscow, Russia; 9Scientific Center for Translation Medicine, Sirius University of Science and Technology, 354340 Sochi, Russia; 10Institute of Molecular Medicine, Sechenov First Moscow State Medical University, 119991 Moscow, Russia; 11Belozersky Institute of Physico-Chemical Biology, Lomonosov Moscow State University, 119992 Moscow, Russia; 12Faculty of Health and Medical Sciences, University of Surrey, Guildford GU2 7X, UK

**Keywords:** Cathepsin K, protease, diagnostics, therapy, inhibitors

## Abstract

Cathepsin K (CatK) is a part of the family of cysteine proteases involved in many important processes, including the degradation activity of collagen 1 and elastin in bone resorption. Changes in levels of CatK are associated with various pathological conditions, primarily related to bone and cartilage degradation, such as pycnodysostosis (associated with CatK deficiency), osteoporosis, and osteoarthritis (associated with CatK overexpression). Recently, the increased secretion of CatK is being highly correlated to vascular inflammation, hypersensitivity pneumonitis, Wegener granulomatosis, berylliosis, tuberculosis, as well as with tumor progression. Due to the wide spectrum of diseases in which CatK is involved, the design and validation of active site-specific inhibitors has been a subject of keen interest in pharmaceutical companies in recent decades. In this review, we summarized the molecular background of CatK and its involvement in various diseases, as well as its clinical significance for diagnosis and therapy.

## 1. Introduction

According to the MEROPS database [1] approximately 7% of the human genome comprises (known and putative) genes translating for 990 proteases and 1605 protease inhibitors [2] like cystatins against cysteine proteases, serpins against serine proteases, and tissue inhibitors of metalloproteases (TIMPs) that regulate proteolytic degradation [3]. Altogether, these elements compose a complex network known as the degradome which represents the set of proteases and inhibitors that characterize a certain environment at any level of organization (cell, tissue, organism, etc.) and at a certain time (homeostasis, pathology, aging) [4,5]. In pathological conditions, the protease–inhibitor–substrate network may be disrupted, resulting in altered protease signaling initiation and transduction, leading to abnormal biological effects and contributing to the development of various inflammatory, neurodegenerative, and cardiovascular diseases, viral infections, atherosclerosis, osteoporosis, and cancer (cancer degradome) [5]. Cathepsins, a group of lysosomal enzymes, are responsible for the degradation of intracellular substrates and crinophagy referred to as the degradation of secretory protein excess stored in secretory granules [6,7]. Lysosomal function is crucial for cancer progression, apoptosis, proliferation, inflammatory cell recruitment, metastasis, chemoresistance, and angiogenesis [8]. Traditionally, cysteine cathepsins were believed to be active only in the acidic pH of the lysosomes; however, it was shown that they are also active at a higher pH, favoring pathologic phenomena such as cancer cell invasion [9].

Cathepsin K (CatK) is a cysteine protease with endo and collagenolytic activities, essential for extracellular type I collagen and elastin matrix recycling, and it is considered the main specific protease of osteoclasts and activated macrophages [10].

Although it has mainly been associated with osteoarthritis, high levels of CatK were found in other pathological conditions, including vascular inflammation, hypersensitivity pneumonitis, sarcoidosis, Wegener granulomatosis, berylliosis, and tuberculosis [11]. Recent studies showed its association with malignancies, further increasing the interest in this enzyme [12].

The designing of specific CatK inhibitors represents one of the most promising approaches to treat conditions characterized by an increased activity of this protease, attracting the attention of pharmaceutical companies over recent decades [13,14].

This review provides insights into the CatK genotype, polymorphisms, molecular structure, and unravels its working mechanism function. In addition, we described different pathological conditions derived from the increased or decreased secretion of CatK and its relevance at the molecular level wherever there is available data. This review also provides an overview of the role of CatK as a therapeutic target and diagnostic biomarker and a detailed guide about progresses in CatK inhibitors development and testing, summarizing the major obstacles to overcome and to regulate the activity of this cysteine protease.

## 2. CatK Structure, Mechanism of Action, Genotype, and Polymorphism

Human CatK is encoded by a single-copy gene of approximately 12.1 kb on chromosome 1q21, with eight exons and seven introns with a similar organization to CatL and CatS [9]. CatK is 329 amino acids long, comprising 15 N-terminal amino acids, 99 amino acids forming the pro-peptides portion, and 215 amino acids forming the catalytic unit. Mature CatK differs from either CatB (24% identity) and CatF (40% identity), but shares approximately 60% of the amino acid identity sequence with CatL, CatS, and CatV, belonging to the CatL-like cluster of the C1A family. These cathepsins share a characteristically shaped active groove conformation, but CatK differs from this cluster for positively charged basic residues favoring the allosteric accommodation of negatively charged glycosaminoglycans (GAGs). This structure permits the formation of high-molecular-weight complexes of the enzyme and the GAGs [15,16].

Two consensus Sp1 binding sites, together with a rich GpC region, were identified as the promoter region of the gene. Several transcription regulatory elements may be proposed as CatK transcription enhancers in different cell types, including activator protein -1 and -3, hepatocyte-acute phase factor-l, Erythroblast Transformation Specific (ETS) putative oncogene Spi-1 (PU1), ETS-1, polyomavirus enhancer activator-3, microphthalmia transcription factor (Mitf), and Transcription Factor Binding To IGHM Enhancer 3 [11]. McQueney et al. showed in vitro that CatK activation is autocatalytic and does not require the activity of a different protease [17].

So far, the mechanism of CatK trafficking in lysosomes has not been fully understood. However, recent studies have suggested a two way-trafficking of the proenzyme. As the proenzyme is tagged by the MP6 (mannose 6-phosphate) receptors in the trans-Golgi network (TGN), it could be either sorted to the lysosomes directly, or reach these organelles after a passage in the plasma membrane. Both routes depend on clathrin-coated vesicles that carry the cathepsins from the TGN or the plasma membrane to the endo/lysosomes [18].

However, in pathological conditions, the synthesis, trafficking, and the levels of the proenzyme in the lysosomes might vary and can be associated with an impaired transcriptional or translational regulation [19]. CatK gene organization, protein structure, expression regulators, and associated disease are shown in Figure 1.

### 2.1. Regulation of CatK Transcription

CatK regulation was deeply investigated in osteoclast differentiation. The process begins with the RANKL binding to its membrane receptor RANK (Figure 2). RANKL is a membrane factor produced by osteoblasts and stromal cells in response to a variety of signals, and many studies showed that it is one of the crucial mediators of CatK regulation [20]. Namely, RANKL was shown to stimulate CatK mRNA and protein expression in human osteoclasts [21], and in murine myeloid RAW 264.7 cells when induced toward an osteoclastic phenotype [22].

Additionally, Pang et al. demonstrated that the RANKL stimulation of CatK mRNA expression occurs in a dose- and time-dependent manner [23].

Many agents positively and negatively regulate the secretion of RANKL in osteoblasts and stromal cells. Stimulators include vitamin D, parathyroid hormone, TNF-α (tumor necrosis factor alpha), glucocorticoids, interleukins 1 (IL-1) and 11 (IL-11), thyroid hormone, prostaglandin E2, lipopolysaccharide, fibroblast growth factor-2, histamine, insulin-like growth factor-1, histamine, and low gravity. On the other hand, inhibitors of RANKL expression include estrogen and transforming growth factor–β, which could lead to menopausal osteoporosis, considering that the process is accompanied by lower levels of estrogen [24,25].

CatK transcription induction by RANKL is stimulated by several mechanisms. An early event in RANKL-mediated signaling involves the activation of TNF receptor-associated factor 6, a critical adaptor molecule for the cognate receptor of RANKL, the overexpression of which stimulates CatK promotor activity. Additionally, RANKL leads to the nuclear factor of activated T-cells (NFAT2) phosphorylation by p38, inducing the translocation of NFAT2 into the nucleus and subsequent transactivation of the human CatK promotor [26]. Mitogen-activated protein kinase kinase 6 overexpression, which enhances p38 activity, also stimulates CatK gene expression and promoter activity. Furthermore, cyclosporine, which inhibits the phosphatase activity of calcineurin, inhibiting NFAT activation, suppresses the stimulation of CatK mRNA expression by RANKL. In addition, inhibitors of calcium signaling, such as BAPTA-AM (Cell-permeable Ca^2+^ chelator) and the calcium ionophore, A23187, reduce RANKL-induced CatK mRNA expression. The RANKL treatment of cells also induces the phosphorylation of Mitf via p38 [27]. Mitf is the mi gene product and is a member of the helix-loop-helix (HLH) leucine zipper family directly regulating CatK gene transcription. Dominant negative mutations of Mitf can cause osteopetrosis as part of the COMMAD syndrome (Coloboma, Osteopetrosis, Microphtalmia, Macrocephaly, Albinism, and deafness) [28,29]. In contrast, in cultured osteoclasts, the overexpression of wild-type Mitf or TFE3, a member of the same transcription factor family, significantly enhanced CatK expression due to the binding of Mitf to three E-box motifs in the human CatK promoter. Moreover, both Mitf and PU.1 synergistically potentiated the NFAT2 stimulation of human CatK promotor activity [26]. Additionally, retinoic acid can stimulate CatK expression in bone physiology [30]. Furthermore, integrin-binding extracellular matrix proteins, such as collagen type I, fibronectin, vitronectin, and osteopontin, can increase CatK mRNA generation [31]. On the other hand, the physiological inhibitors of osteoclast differentiation and activation, such as osteoprotegerin (OPG), can also directly suppress CatK expression [32]. Among the inhibitors of CatK mRNA expression are estrogen [32], Interferon-γ (INF-γ), IL-10, -12, and -18 while IL-1α, -6, -11, -15, and -17 stimulate CatK expression in osteoclasts [33,34,35]. The negative effect of estrogens on osteoclasts is mediated by a membrane receptor inducing a phosphorylation cascade of various proteins, including Src. It was shown that calcitonin suppresses CatK expression at the mRNA level, while the parathyroid hormone increases it [36].

CatK activity and stability in the extracellular space have been shown to be regulated by GAGs that interact electrostatically with this enzyme [37].

On the other hand, proteins such as Clusterin can bind and stabilize CatK at neutral pH and decrease CatK inhibition by substrate excess [38].

CatK can cleave many proteins, including elastin, gelatin, osteopontin, osteonectin [39,40] collagen [13], and aggrecan, both in the interglobular and in the globular domain attached to the hyaluronan [41]. A characteristic difference of CatK from other proteases only breaking down collagen telopeptides is that it cleaves 3-helical collagen. It was reported that CatK works more effectively in combination with the major GAGs chondroitin sulfate [42]. In addition, CatK can perform intracollagen helix proteolysis similar to metalloproteases [43]. All this evidence makes CatK a unique collagenase.

In actively bone-resorbing osteoclasts, vesicles with CatK are transported to the prelacunar space, and CatK is secreted into the resorption lacuna, where it lyses the bone matrix. Then, CatK and the matrix degradation products undergo endocytosis. Subsequently, vesicles containing monomeric bone TRAP (thrombospondin-related anonymous protein) merge with endocytic vesicles. CatK activates bone TRAP, which generates reactive oxygen species, thus completing the degradation of the matrix components [44].

## 3. Role of CatK in Diseases

### 3.1. Bone

Besides osteoclasts, CatK is expressed in pre-osteoclasts, bronchial epithelium, bile ducts, chondrocytes [45], and smooth muscle cells of arteries affected by atherosclerosis [46]. It has been shown that a defect in the CatK propeptide and mutations in the polypeptide chain of the mature form of the enzyme prevent its correct folding, leading to its inability to bind and break down the collagen. This phenomenon results in pycnodysostosis [47], a skeletal dysplasia characterized by impaired bone remodeling leading to osteosclerosis, frequent fractures, clavicle dysplasia, and acro-osteolysis of the distal phalanges [48]. Acro-osteolysis and high bone density revealed by radiological assessment is a pathognomonic symptom for this disorder. However, confirmation of this diagnosis can be provided by CatK molecular analysis which shows pathologic alterations in both the alleles. Mutations in this gene were also detected in patients with diagnosed osteopetrosis. It is crucial to distinguish both disorders due to the successful treatment of osteopetrosis by hematopoietic stem cell transplantation at early stages [49]. Recent studies suggest that CatK deficiency is involved in promoting alveolar bone regeneration mediated by jawbone marrow mesenchymal stem cells (JBMMSCs). This finding is supported by RNAseq data, suggesting that CatK inhibition or knockout could promote alveolar bone regeneration by activation of JBMMSCs and enzymes, which is significantly up-regulated and uses glycolysis pathway [50]. High levels of CatK are related to osteoarthritis. Due to its collagenolytic activity, it was speculated that CatK may be involved in early cartilage degradation, increased pathological angiogenesis, and recruitment of osteoclasts, further accelerating this degenerative process [51]. Khoshdel et al. demonstrated high CatK serum levels in patients with osteoarthritis compared to healthy ones; however, there was no correlation between CatK levels and course of the disease [52]. CatK being highly expressed in both osteoclasts and chondrocytes during osteoarthritis progression, it is still unclear which cell population is the key player in this mechanism [53]. Other studies suggest that elevated CatK levels are associated with postmenopausal osteoporosis [54]. Due to similarities with osteoporosis, it was speculated that CatK could play a crucial role in bone mineralization, particularly in patients undergoing chronic hemodialysis, wherein elevated parathyroid hormone levels can induce CatK expression. For this reason, CatK was suggested as a biomarker of parathyroid hormone disorders [55]. Recent studies suggest that Gaucher disease (GD) as an inherited metabolic disorder that leads to liver, spleen, and bone abnormalities can also be associated with increased levels of CatK. Observed elevated CatK levels in GD patients may lead to osteoporosis and lytic bone lesions [56]. Currently, the role of CatK in anti-osteoporosis treatment is of essential meaning and the understanding of its biology represents a promising way to understand its role outside bone. For example, the intracellular collagen degradation after endocytosis can reveal its role as a potential tumorigenic activator [57].

### 3.2. Heart and Vascular System

CatK is involved in ischemia-induced neovascularization. Due to its overexpression, c-Notch1 (Notch homolog 1, translocation-associated) levels and its downstream signaling pathways are enhanced, suggesting that CatK could contribute to Notch1-dependent neovascularization and be a potential therapeutic target for ischemic disease [58]. However, there are suggestions that CatK deficiency may aggravate ischemia-induced neovascularization associated with the decrease in c-Notch-Akt signaling activation. In particular, bone marrow-derived c-Kit+ cells of aged CatK−/− mice have lower levels of c-Notch1 and p-Akt proteins compared to the control cells of CatK+/+ mice [59]. An increased level of CatK was found in patients with ischemic heart disease. High blood levels of CatK could represent an independent predictor and novel biomarker for the diagnosis of coronary heart disease [60]. In addition, it was shown that CatK knockout attenuated cardiac oxidative stress and calcineurin/NFAT signaling in mice with STZ-induced diabetes. In cultured H9c2 myoblasts, CatK inhibition protected the cells from oxidative stress and apoptosis induced by high glucose levels, showing that CatK may represent a potential target for treating diabetes-related cardiac dysfunction [61]. Besides the increased level of CatK, mislocalization and inappropriate secretion could trigger cardiovascular defects. Namely, in zebrafish models of mucopolipidosis II (lysosomal disorder caused by mutations in GlcNAc-1-phosphotransferase enzyme), lysosomal hydrolases, including CatK, lack appropriate secretion, causing valvular heart disease and myocardial formation [62].

In chronic kidney disease, circulating CatK levels were significantly higher in patients with calcification of the coronary arteries, representing a potential biomarker of the disease severity [63].

It was reported that circulating miR-185-5p in plasma is reduced in acute coronary syndrome patients. The endogenous inhibition of miR-185-5p in endothelial cells promotes angiogenesis, resulting in an accelerated repair of heart function after MI through CatK gene expression targeting. Therefore, the modulation of CatK gene expression by miR-185-5p could be of a significant importance for a future angiogenic therapy in treating ischemic diseases, such as MI, stroke, peripheral artery disease, and wound healing in diabetes [64].

### 3.3. Immune System and Lungs

Recent studies suggested that even though CatK does not have antigen processing properties, it can mediate the immune response against foreign DNA through the activation of troll-like receptor-9, expressed by CD4 cells, the main mediators of autoimmune diseases [65], including psoriasis. In lungs, increased CatK activity is pivotal for the maintenance of collagen matrix homeostasis and to contrast its excessive deposition by activated pulmonary lung fibroblasts. The study was obtained comparing CATK−/− with CATK+/+ control mice and also showed an increased level of CatK activity in specimens from lung fibroblasts, compared with specimens from normal lungs, suggesting the importance of CatK activity in lung fibrosis [10].

### 3.4. Central Nervous System (CNS)

Increased activity of CatK has been found in various CNS diseases, including stroke, cerebral aneurysm (CA), chronic subdural hematoma, and schizophrenia. Interestingly, some studies suggest a protective role of CatK in acute ischemic stroke [66]. Aoki et al. showed that the imbalance between cathepsins, including CatK, and their inhibitors might lead to an excessive disruption of the extracellular matrix in arterial walls, leading to the progression of CAs [67]. Bernstein and Lendeckel showed that CatK downregulation is associated with the assumption of amphetamines, but is upregulated by typical and atypical neuroleptics. Namely, in studies on postmortem brains of chronic schizophrenics, they found a significant increase in CatK protein expression in the CNS of patients receiving long-term treatment with neuroleptics [68,69]. What gives CatK even more importance is its association with metabolic disorders resulting from long-term neuroleptic therapy in people suffering from schizophrenia [70]. The neuroleptic treatments of schizophrenia can affect many tissues, including adipose tissue. Upregulated CatK in lipid and glucose metabolism, together with the negative effects of neuroleptic treatment on adipose tissue, can lead to deviations and metabolic complications. Thus, by inhibiting CatK, the normalization of the lipid and glucose metabolism in people suffering from schizophrenia can be achieved [68]. The molecular mechanism of CatK’s involvement in CNS diseases is yet to be explained.

### 3.5. Skin

Recently, CatK was recognized as a possible target in psoriasis treatment. Increased levels of CatK mRNA were documented in psoriasis-like lesions in the epidermis and dermis of K5.Stat3C mice compared with wild-type mice. The same research showed that the CatK inhibitor NC-2300 (also known as VEL-0230) ameliorates TPA (12-O-Tetradecanoylphorbol-13-acetate) -induced psoriasis-like lesions. NC-2300 is a potent CatK inhibitor with dual-acting properties that both stimulate bone formation and inhibit its loss. NC-2300 is being studied preclinically to treat diseases involving bone mineral disorders such as bone loss related to multiple myeloma, osteoporosis, bone metastases, and rheumatoid arthritis. Insights in psoriasis lesion development show that CatK is involved throughout the TLR7-mediated IL-23/Th17 pathway [71].

### 3.6. Cancer

Being mainly associated with inflammatory diseases [11], high levels of CatK were also found in different tumors [72]. The mechanism of CatK involvement in tumor cells proliferation may be related to RANK/RANKL, TGF-β, mTOR, and the Wnt/β-catenin signaling pathway [72,73].

Its expression has been observed in stromal and in cancer cells [12], including tumor-associated fibroblasts and macrophages in invasive adenocarcinomas [74], renal and extrarenal perivascular epithelioid tumors (PEComas), alveolar soft part sarcoma, osteolytic lesions of giant cells in bone tumors, chondrosarcoma [75,76,77], basal cell carcinoma, oral tongue squamous carcinoma cells [78,79,80] breast and prostate cancer (PCa), and other epithelial-derived cell cancers [81,82].

The expression of a calcium-sensing receptor (CaSR) in patients with adenocarcinoma leads to higher CatK levels and hormone calcitonin receptors in osteoclasts, which are related to bone matrix degradation [83].

CatK expression in chordoma patients occurs during neoplastic transformation and does not appear in chorda dorsalis, while in chondrosarcoma CatK expression is limited to osteoclastic cells localized between infiltrative tumor components and regular bone trabeculae [84].

CatK is suggested to be a powerful diagnostic marker for a wide spectrum of perivascular epithelioid tumors (PEComas), as well as possible differentiator of PEComas from other human cancers [77].

Alveolar soft part sarcoma is characterized by a specific chromosomal alteration, der(17)t(X:17)(p11:q25), resulting in fusion of TFE3 with alveolar soft part sarcoma critical region 1 (ASPSCR1) at 17q25 [85]. As we mentioned before (in Section 2.1. Regulation of CatK transcription), TFE3 overexpression significantly enhanced CatK expression due to the binding to three E-box motifs in the human CatK promoter. 

The basal cell carcinoma lesions show the overexpression of TGF-β, Smad2, cathepsin-K, and MMP-1, -3, -8, and -9 when compared to healthy control skin samples [86]. It is suggested that CatK mediates intracellular degradation of matrix proteins after phagocytosis, resulting in invasion and metastasis [57].

CatK was reported to be overexpressed in oral tongue squamous cell carcinoma tumor epithelial cells, but not in normal mucosa keratinocyte. The organization of the lymph node stroma expresses increased amounts of CatK. Investigations of CtaK levels in the tumor microenvironment suggested its potential role as prognostic parameter [79].

Reithmeier et al. showed that CatK colocalized with tartrate-resistant acid phosphatase (TRAP) and was involved in regulating the secretion of TRAP 5a in a breast cancer cell line, although the connection between CatK and TRAP 5a processing to TRAP 5b isoform was not found to be essential [87]. Other studies showed that CatK induced platelet aggregation by the up-regulation of sonic hedgehog (SHH), parathyroid hormone-related protein (PTHrP), osteopontin (OPN), and TGFβ in epithelial–mesenchymal-like cells from patients with Luminal B breast cancer [88].

Latest research revealed a molecular mechanism connecting the increase in CatK levels to tumor growth and metastasis in castration-resistant prostate cancer. A strong correlation between CatK and IL-17 signaling pathways, as well as a high concentration of M2 tumor-associated macrophages (TAMs) M2, has been reported [89].

Particularly in breast and prostate cancer, CatK activity was significantly increased in bone metastases, rather than the corresponding primary tumors, probably due to its involvement in bone resorption, facilitating tumor metastatic spreading [85]. In a cohort study for the expression of tumor biomarkers including CatK and MMP9 on non-functioning pituitary adenomas with different invasion patterns, it was shown that an increased expression of CatK was related to sphenoid sinus, while an increased expression of MMP9 and MMP2 was related to cavernous sinus [90].

Some of the inflammatory mediators leading to tumor progression and metastases are bone marrow macrophages (BMMs), which together with osteoclasts share common hematopoietic progenitors [91] and express CatK in bone tumors [92]. Previous studies suggested that CatK overexpression together with the secretion of pro-inflammatory cytokines (IL-6, -8) is one of the key players in metastatic tumor progression in the bone [77]. Herroon et al. demonstrated reduced prostate tumor progression in CatK knockout mice. This is explained by the reduction in proinflamatory signals exchanged between BMM and osteoclasts in CatK deficiency [93].

A detailed list of diseases related to CatK activity is shown in Table 1. Many of them are associated with an increased expression and secretion of CatK. A potential working mechanism of this phenomenon is shown in Figure 3.

## 4. CatK Inhibitors for Bone and Other Diseases

The design of active site-specific inhibitors for human CatK has been a subject of keen interest in pharmaceutical companies over recent decades [108]. Recent studies based on computational approaches of molecular docking methods showed that this could represent a fast and reliable way to identify CatK ectosteric inhibitors [109]. Many synthetic compounds and natural products can be used to develop CatK inhibitors. Irreversible covalent inhibitors showed antigenic and immunologic complications, leading to pycnodysostosis in humans and osteopetrosis in mouse models [13]. On the other hand, reversible covalent and non-covalent inhibitors could cause off-target effects despite excellent biochemical selectivity [110]. Today, most of the research is focused on the creation of the cysteine-thiol component of CatK with reactive electrophilic “warheads” for the reversible or irreversible inhibition of its proteolytic activity [36].

Research into CatK as a new target for osteoclast-related diseases has focused on screening and developing natural and synthetic inhibitors to treat diseases such as osteoporosis, osteoarthritis, and other bone disorders, characterized by excessive resorption levels [13]. A recent clinical trial reports MIV-711 as a potent and selective CatK inhibitor with dose-dependent effects improving bone and cartilage degradation in monkeys and humans [76]. CatK inhibition provided significant advantages to protect the integrity of the subchondral bone and cartilage and to reduce osteophytosis in animal models of osteoarthritis [51]. Another project for the development of a synthetic CatK inhibitor, ONO-5334, from Ono Pharmaceutical Co, which was in phase I and II clinical trials, was also closed for market reasons [36]. A new highly effective and selective inhibitor of CatK, MIV-711, showed clinical potential to treat bone and cartilage-related diseases in humans, such as osteoarthritis, as confirmed in phase II clinical trials [111].

Even though currently there are no FDA-approved drugs, several generations of CatK inhibitors have been developed, such as relacatib, balicatib, and odanacatib [14]. A study of relacatib against the background of 9-month therapy in monkeys with ovariectomy showed a dose-dependent decrease in the bone resorption parameters serum C-terminal crosslinking telopeptides of type I collagen (sCTx) and urinary N-terminal crosslinking telopeptides of type I collagen (uNTx) concentrations with a preserved level of BMD [112]. Similar results were obtained in an 18-month study of balicatib [113]. In both studies, histomorphometric analysis showed that these drugs reduced bone resorption rates in the trabecular and cortical regions of the bone. A more potent and selective CatK inhibitor, odanacatib, demonstrated not only a suppression of the bone resorption markers sCTx and uNTx, but also an increase in the spinal bone mineral density when administered to ovariectomized rabbits and monkeys [114]. Recently, it was reported that odanacatib inhibited the adhesion and signaling pathway for MMP-9, PI3K, and MAPK of the highly metastatic human breast cancer cell line MDA-MB-231. These results indicate the high therapeutic potential of odanacatib in breast cancer patients [115]. Furthermore, a recent study demonstrates that the inhibition and knockdown of CatK by odanacatib (ODN) enhanced apopotosis in several cancer cell lines induced by oxaliplatin [116].

However, clinical trials of nonselective drugs in this group were affected by emerging side effects. A relacatib study was stopped after phase I due to its interaction with commonly prescribed drugs (i.e., acetaminophen, ibuprofen, and atorvastatin) [117], while a balicatib study was discontinued due to rash and sclero-like skin lesion appearance [117]. For odanacatib alone, randomized controlled trials (RCTs) have been conducted in women with postmenopausal osteoporosis in order to assess its effectiveness in reducing the risk of fractures. In phase II studies, odanacatib significantly increased BMD in clinically significant areas of the skeleton and decreased the concentration of bone resorption markers (sCTx, uNTx) [118]. At the same time, the levels of bone formation markers, such as bone-specific alkaline phosphatase and the amino-terminal propeptide of type I procollagen (P1NP), declined after the first months of therapy, indicating that this drug mainly suppressed bone resorption, rather than increasing bone formation [119]. In the axial placebo-controlled phase III trial LOFT (Long-term Odanacatib Fracture Trial) involving 16,713 women with postmenopausal osteoporosis, assessing the effectiveness of the drug in reducing the risk of fractures, an increased risk of cardiovascular events (cerebrovascular accident and myocardial infarction) was observed. It is currently unclear whether this was due to the weakening of the non-skeletal degradation of CatK or off-target inhibition. However, these data prevented any further FDA approval [120]. In another study in which odanacatib was administered to women with breast cancer and bone metastases was withdrawn in Phase 3 due to toxicity (NCT00692458), regardless of the proven success in the earlier phases of the study [121]. The novel water-soluble artemisinin analogue, SM934, combined with testosterone, could suppress the viability, proliferation, and metastasis of human breast cancer cells (MDA-MB-231 and SKBR-3) by inhibiting CatK expression and consequently decreasing Bcl-xL activity, which is an anti-apoptotic protein involved in metastasis [122].

Recent data suggest new promising highly potent inhibitors of CatK based on cyanohydrazide warheads. This study investigated the effect of two different positions of the warhead: an azadipeptide nitrile and 3-cyano-3-aza-β-amino acid. The binding with mature CatK and with intermediate CatK has shown the picomolar potency of 3-cyano-3-aza-β-amino acid. Furthermore, by live cell imaging, it was demonstrated that cyanohydrazides successfully target CatK in osteosarcoma cells, suppressing its development [123].

Adeno-associated virus (AAV)-mediated CatK silencing was proven to be a successful treatment for endodontial [124] and periodontal [125,126] diseases. Specifically, it was shown that local (AAV)-mediated RNAi silencing of CatK drastically reduced inflammation by lowering the expression of many inflammatory cytokines, T cell, and dendritic cell infiltration in periodontal lesions in mice [125]. Similar results show that AAV-RNAi CatK silencing efficiently reduced osteoclast bone resorption and lesion size in vitro [124].

Finally, a systemic inhibition of CatK has been warned to possibly increase the risk of lung fibrosis [10] and make skin more prone to excessive matrix protein deposition in response to trauma [127].

## 5. Experimental Perspectives of CatK in Disease Diagnosis and Therapy

CatK may be used as a diagnostic biomarker. Its activity in osteoclasts may be used for in vivo imaging of osteoclasts, providing a means for the early diagnosis of upregulated resorption and rapid feedback on the efficacy of the treatment protocols prior to significant loss of bone in the patient. The near-infrared CatK probe comprised MPEG d-poly-lysine amino acid backbone chain functionalized with Cy5.5 fluorophore through the CatK-sensitive link sequence and was activated when cleaved by CatK [128].

Furthermore, a recent work showed a diagnostic platform for epithelial ovarian cancer based on molecular, biophysical, and electrochemical research in silico and in vitro. The platform is based on the specific interaction between cystatin C as a ligand and CatK and cathepsin CatL as biomarkers, representing a promising tool for diagnostics and the timely detection of epithelial ovarian cancer [129].

Recent studies suggest a novel therapy for inflammatory arthritis based on a stimuli-responsive drug delivery system [130].

Additionally, CatK immunostaining is generally used to detect TFE3/TFEB rearrangement in renal cell carcinomas [19]. High CatK levels are significantly associated with mineral dysmetabolism and inflammation, and can predict cardiovascular death in end-stage kidney disease (ESKD) patients [131]. The diagnostic role of CatK in predicting major adverse cardiac and cerebrovascular events has been shown to be independent from the predicting role of brain natriuretic peptides. Elevated CatK levels are significantly associated with coronary artery calcification and can represent an independent predictor of major adverse cardiac and cerebrovascular events in patients with non-diabetic chronic kidney disease [132]. CatK increase in primary prostate tumor and in skeletal metastatic tumor represents a disease progression indicator. Studies on advanced-stage PCa in murine bones have shown that CatK inhibition could prevent the establishment of skeletal PCa and retard tumor progression [40]. Siddiqui et al. demonstrated that PCa-secreted growth differentiation factor-15 (GDF15) promotes bone metastases and induces bone micro-architectural alterations. This study showed that GDF15 enhances osteoblast function and facilitates CatK expression, which provides a niche for PCa growth [133]. There are indications of the potential use of some phytocompounds that are proven CatK inhibitors, in the prevention of coronavirus cellular entry and replication. A recent study predicted the inhibitory potential of curcumin, allicin, and gingerol towards CatK, SARS-CoV-2 main protease, and SARS-CoV-3 C-like protease [134]. These results could indicate some potential in the use of synthetic CatK inhibitors in COVID-19 treatment.

## 6. Conclusions

CatK is a highly active protease and has many important roles in many tissues and organs. Although the main signaling pathway regulator is RANKL/RANK, many other factors (e.g., hormones, cytokines, etc.) can affect the expression of CatK. In this context, more investigations are needed, in particular in tissue different from bones, where its regulation mechanism was extensively investigated. More information is also needed to understand the biology of these proteases in different diseases such as CNS and skin disorders. In this context, the use of disease model knock-outs for this protease could reveal its role in different organs and pathologies. On the other hand, research in developing new inhibitors and diagnostic tools revealing CatK activity is pivotal in order to develop new therapeutic strategies and understanding the biology of this pleiotropic protease.

## Figures and Tables

**Figure 1 ijms-23-13762-f001:**
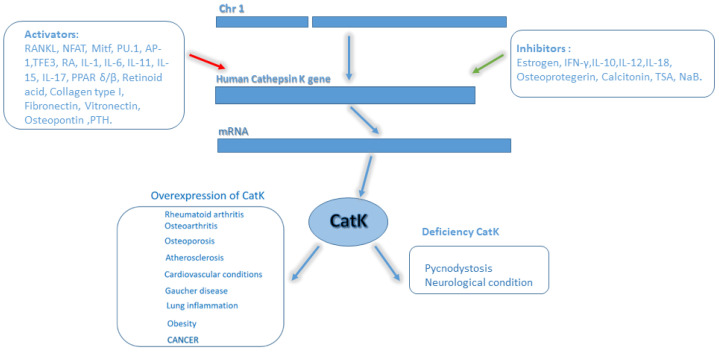
Localization and regulation of CatK gene and involvement in pathological conditions. Human CatK is encoded by a single-copy gene of approximately 12.1 kb on chromosome 1q21, with 8 exons and 7 introns. CatK is 329 amino acids long, comprising signal peptide 15 N-terminal amino acids, 99 forming the pro-peptide, and 215 amino acids forming the catalytic unit. CatK expression is regulated by multiple proteins in positive and negative manners. Overexpression of CatK causes picnodystosis and several neurological conditions.

**Figure 2 ijms-23-13762-f002:**
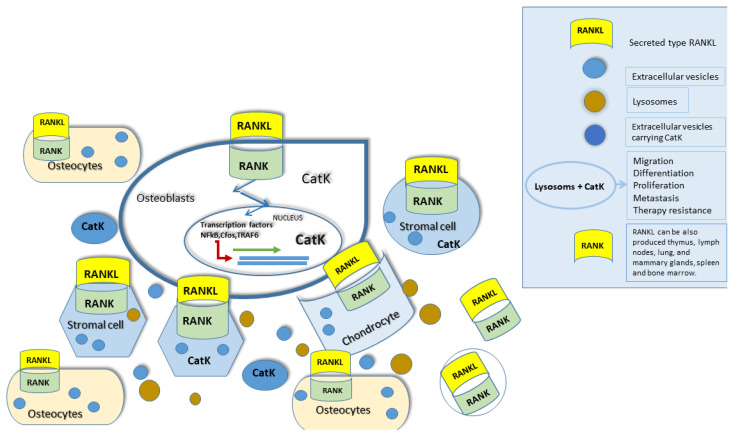
RANK/RANKL signaling. RANKL (Receptor activator of nuclear factor-kappa B ligand) to this date, represents the only known physiological agonist for RANK, its receptor. RANK does not exhibit any kinase activity. Signaling pathway of RANKL/RANK is crucial for skeletal homeostasis and its misbalance affects bone resorption. The outcome of disturbed skeletal homeostasis is the cause for a number of bone diseases, e.g., osteoporosis, rheumatoid arthritis, and osteopetrosis. RANKL/RANK signaling also plays an essential role in lymph node development and in mammary glands in mice, as well as thymic maturation.

**Figure 3 ijms-23-13762-f003:**
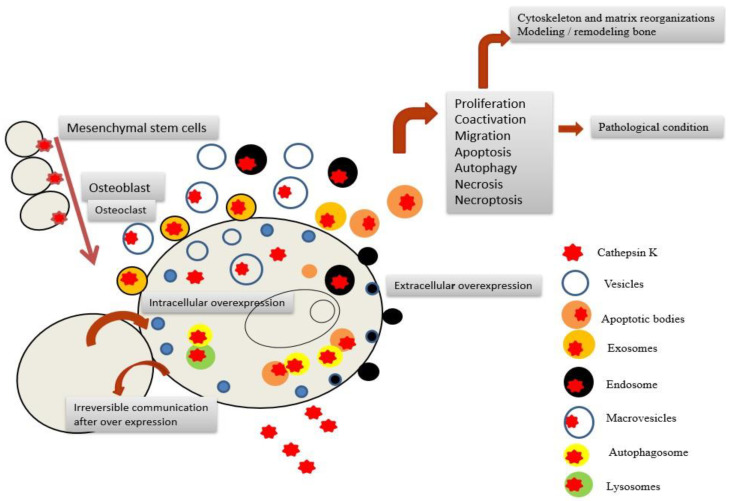
Schematic of the potential involvement of CatK in disease.

**Table 1 ijms-23-13762-t001:** Involvement of CatK in different diseases classified by tissue.

Tissue	Disease	Involvement of CatK
Bone	Pycnodysostosis (PD)	A homozygous or compound heterozygous mutation (mutations that occur on different copies of genes and may completely “knock-out” gene function) in the CatK gene 236G > A, 121–1G > A and 926T > C, causes lower degradation of type I collagen due to lack of CatK activity [94,95].
	Postmenopausal osteoporosis (PO)	In a combination of hyper-active osteoclasts and less functional osteoblasts (typically for estrogen-deficient women), CatK inhibitors can target the resorption process [54].
	Osteoarthritis (OA)	The up-regulated expression of the CatK collagenase activity could affect the cartilage matrix degradation in the late stadium of osteoarthritis [96].
	Rheumatoid arthritis (RA)	CatK inhibition can play a significant role in cartilage degradation, retarding the bone loss process and joint destruction [97,98,99].
	Gaucher Disease (GD)	A multisystemic disorder, associated with progressive accumulation of Gaucher cells—large macrophages that store glucocerebroside. In GD patients elevated CatK levels are observed, which can lead to osteoporosis and lytic bone lesions [56].
Skin	Psoriasis (PS)	Psoriatic lesions (an increased copy number of variations in β-defensin gene locus) express an elevated level of CatK compared to healthy skin [65].
	Scar formation (SF)	In surgical scars compared to normal skin, is observed a proteolytic activity of CatK [57].
Lung	Lung fibrosis (LF)	CatK protects against matrix deposition in bleomycin-induced LF [100].
	Pulmonary lymphangioleiomyomatosis (PLAM)	General expression of CatK activity was observed during PLAM [101,102].
	Atherosclerosis (AS)	Stress-induced—CatK expression in endothelial cells can lead to vascular remodeling and atherosclerosis [103].
Adipose tissue	Obesity/overweight	CatK expression can be detected in pre-adipocytes and be further up-regulated in the process of differentiation. The lack of CatK retards the adipogenesis [104,105].
Central nervous system	Schizophrenia	High levels of CatK in individuals suffering from schizophrenia, as a result of long-term treatment of the patients with neuroleptics [68,106].
	Cerebral aneurysm (CA)	Cystein cathepsins cause degradation of ECM in aneurysmal wall in the late state of CA. The administration of cysteine proteases inhibitors (CPIs) leads to prevention of CA progression [67].
	Chronic subdural hematoma (CSH)	The expression of CatK in CSH patients, may lead to CSH development [107].
Cancer	Adenocarcinoma Chondrosarcoma Renal and extrarenal perivascular epithelioid tumors (PEComas) Alveolar soft part sarcoma Basal cell carcinoma Oral tongue squamous carcinoma cells Breast and prostate cancer Other epithelial-derived cell cancers	High levels of cathepsin K have been observed in metastatic tumors, which correlates with its primary function to degrade collagen, thus aiding tumor invasion and metastasis [82].

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
