# Peer review of "Cathepsin K in Pathological Conditions and New Therapeutic and Diagnostic Perspectives"

_ijms, 2022, doi:10.3390/ijms232213762_

Round 1
Reviewer 1 Report
The review by Mijanović et al. of insights into Cathepsin K from the molecular level to disease is very well organized and deserves to be published in the International Journal of Molecular Sciences. My only complaint in the current manuscript is that after 3.3, there is less description of CatK's relevance to disease at the molecular level. Although this may be due to the lack of information on these matters, if possible, I hope these will be added.
Author Response
The review by Mijanović et al. of insights into Cathepsin K from the molecular level to disease is very well organized and deserves to be published in the International Journal of Molecular Sciences. My only complaint in the current manuscript is that after 3.3, there is less description of CatK's relevance to disease at the molecular level. Although this may be due to the lack of information on these matters, if possible, I hope these will be added.
Answer:
We thank the reviewer for her/his constructive criticism of our review paper. We hope that the answers and the way we extended chapter 3, especially the subtitle 3.5 Cancer, will meet with your approval.
However, molecular mechanisms of CatK involvement in CNS diseases is yet to be discovered and for psoriasis we offered as many information as there is available. These issues were highlighted in the conclusions.
As for role of CatK in various cancers we implemented the description of its function in metastases development. The additional part is highlighted.
Reviewer 2 Report
This is a condensed manuscript in which the role of cathepsin K in physiology and pathology is extensively discussed. However, a similar review article has been published. I suggest that the authors add some contents in this manuscript as below.
-
The authors mention that both overexpression and suppression of CatK are involved in the pathogenesis of diseases. However, neither the text nor the figures figure 1 illustrates the clear mechanisms underlying the effect of CatK. In terms of the action of any protease , the extracellular and intracellular effects of the enzyme are very different. I suggest that the authors should draw a figure to summarize the underlying mechanisms of overexpression and suppression of CatK.
-
A similar review article has already been published. Although the authors have cited this article (ref. 37), they did not highlight Ref 37 as a review article. I strongly suggest that the authors should make a brief introduction of cited review articles first in this manuscript. In addition, authors should mention how their manuscript differs from the topics and contents from other reviewer articles of cathepsin K.
[37] Dai, R.; Wu, Z.; Chu, H.; Lu, J.; Lyi, A.; Liu, J.; Zhang, G. CatK: The actionin and beyond bone. 2020. Frontiers in Cell and 522 Developmental Biology. 8(433).
Author Response
This is a condensed manuscript in which the role of cathepsin K in physiology and pathology is extensively discussed. However, a similar review article has been published. I suggest that the authors add some contents in this manuscript as below.
- The authors mention that both overexpression and suppression of CatK are involved in the pathogenesis of diseases. However, neither the text nor the figures figure 1 illustrates the clear mechanisms underlying the effect of CatK. In terms of the action of any protease, the extracellular and intracellular effects of the enzyme are very different. I suggest that the authors should draw a figure to summarize the underlying mechanisms of overexpression and suppression of CatK.
Answer
We thank the reviewer for her/his constructive criticism of our review paper. Many papers describe an association of CatK expression with the disease, not solid proves of the its mechanism. Also, CatK is involved in many diseases, and drawing a representative picture of the mechanism of action might be difficult. Finally, there are only a few examples of CatK suppression correlating with some diseases. In vision of this, we included one more picture (Figure 3) depicting a potential working mechanism to provide at least an example of the action of CatK in pathologic conditions.
- A similar review article has already been published. Although the authors have cited this article (ref. 37), they did not highlight Ref 37 as a review article. I strongly suggest that the authors should make a brief introduction of cited review articles first in this manuscript. In addition, authors should mention how their manuscript differs from the topics and contents from other reviewer articles of cathepsin K.
Answer
The novelty of this review (compared to ref 37) resides in a detailed explanation of CatK transcription regulation. Also, this review summarizes the role of CatK in the occurrence of metastases in various cancers. Our paper gives detailed guide of progresses in CatK inhibitors testing, and summarizes all the obstacles that future researchers need to overcome to generate more efficient drug. Finally, we offer experimental perspectives of CatK in disease diagnosis and therapy. These differences were highlighted in the last paragraph of the Introduction.